# HloEnv: A Graph Rewrite Environment for Deep Learning Compiler Optimization Research

## Abstract

We introduce HloEnv, an environment based on Accelerated Linear Algebra (XLA) for deep learning (DL) compiler optimization research. HloEnv transforms all graph rewrites into a common representation, providing a flexible interface to control and modify existing graph optimization passes. In this representation, an XLA pass is converted into a set of sequential rewrite decisions, which control when and if the rewrites are applied. Along with HloEnv, we present a dataset with broad coverage of computation graphs drawn from modern real-world machine learning models. We select two XLA passes with the largest impact on the runtime of the compiled program, and explore the potential for further improvement over XLA in this decision space. We show that using simple heuristics for decision-making can achieve on-par or better performance than XLA. Using search algorithms further boosts performance. We intend for HloEnv and our dataset to be an open-source, community-driven effort that helps spur advances in DL compiler optimization research.

## 1 Introduction

Deep Learning (DL) models have been getting significantly larger and more computationally expensive (Thompson et al., 2020). As a result, computational efficiency is now increasingly important for the economic and technical viability, as well as the environmental sustainability of a DL project. DL compiler optimization is important for achieving this efficiency. A DL compiler parses user-defined DL model code (usually written in Python) into a high-level directed acyclic graph (DAG) that can then be optimized to run efficiently on DL hardware through a sequence of sub-graph rewrite passes. Current production-ready DL compilers are still heavily hand-engineered, and require deep domain knowledge to create well-optimized results.

Great efforts have been made to alleviate the reliance on human engineers. TASO (Jia et al., 2019c) is the most representative work on search-based DL compiler optimization. It automatically generates graph rewrites and searches for better optimization solutions on a larger search space. However, the set of DL operators it considers contains only 12 operators, which does not generalize well to newly emerged DL models. Recent works on learning-based DL compiler optimization such as REGAL (Paliwal et al., 2020) and GO (Zhou et al., 2020a) model a limited set of passes each with a different representation. To our best knowledge, there has been no work that generalizes to all optimization passes with a common representation.

In short, at the current stage, research on DL compiler optimization is still facing the following challenges: First, due to their non-unified implementations, there is no systematic interface that has a wide coverage of optimization types. Second, most existing works focus on specific sets of passes. Third, current DL compiler optimization benchmarks use either closed-source or small datasets with a limited set of DL models. The community has not yet centered its efforts to build a publicly accessible dataset of real-world DL computation graphs.

We propose the following to address these challenges. First, we develop HloEnv, an environment for the optimization agent to inter-operate XLA (Leary & Wang, 2017), a production-quality cross-framework DL compiler. This environment provides a common representation for any type of graph rewrites. Second, we present a dataset with broad coverage of High-Level Operations (HLO) graphs drawn from real-world JAX-implemented machine learning code, extracted from a variety of open-source repositories on GitHub (Table A.2), with spectrum spans through various domains. This

provides a more representative dataset of workloads for DL compiler optimization research. Third, based on a thorough analysis of XLA optimization passes, we determine two XLA passes with the most significant impact on the runtime of the compiled program. We explore using simple heuristics and search-based algorithms to further optimize these passes.

The design of HloEnv points to a potential future where DL compiler engineers only need to develop and maintain a simple set of rewrite rules and leave the complicated heuristics to machine learning-generated optimization strategies that generalize to both new DL models and new DL hardware.

## 2 SYSTEM DESIGN OF HLOENV

### 2.1 XLA PRELIMINARIES

XLA compiles computation graphs in High-Level Operations (HLO) IR format into machine instructions for different backends. As part of this compilation process, XLA runs a series of passes to modify the HLO graph. The passes perform rewrites (using pattern matching and replacement) on the HLO graph to optimize the performance or ensure the correctness of the graph. These passes can be composed in a pipeline and recursively grouped in a parent pipeline. These passes/pipelines are run sequentially in a fixed order and can be run either once or repeatedly in a loop until the pass no longer changes the HLO graph.

### 2.2 OVERVIEW OF HLOENV

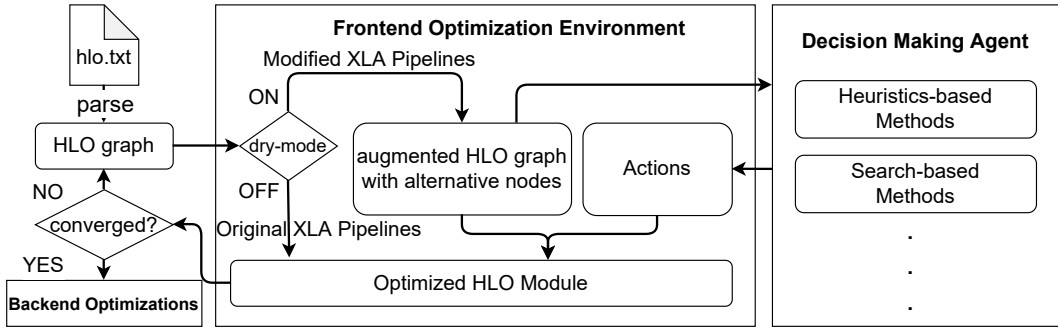

Figure 1: The HloEnv interaction loop.

HloEnv aims to provide a flexible interface that allows for easy control of the XLA optimization passes and pipelines. Each pass and pipeline in HloEnv can be individually set to *dry-mode* to allow us to intercept and control the rewrites they perform.

As shown in Fig. 1, HloEnv's Python interface parses an HLO text file into an HLO graph and loads it into the frontend optimization environment. A user-specified set of XLA passes/pipelines is then applied to the HLO graph. HloEnv executes the original pass/pipeline directly if dry-mode is turned off, while it captures these rewrites without actually applying to the source graph when dry-mode is turned on. An augmented graph that contains both the source graph and all the rewrite opportunities is generated for the user. Using the augmented graph as an input, the user can develop various decision-making agents to decide which rewrites to apply. This process can be applied multiple times until the output HLO graphs stay unchanged (converge) or until a decision is made to end the optimization for that pass/pipeline. The user can then use XLA's backend APIs to generate the final device code.

From the decision-making and control point of view, our system defines a Markov Decision Process (MDP) $\mathcal{M} = (\mathcal{S}, \mathcal{A}, P, R)$. $\mathcal{S}$ stands for the state space, in our case, the augmented graph. From the state, the agent computes the action in the action space $\mathcal{A}$ that decides which rewrite rules to apply. $P$ describes the transition function of the HloEnv, i.e., change of the graph when certain rewrite rules are applied. $R$ is the reward generated from the decision, in our case, the improvement of runtime between the old and new graphs.

HloEnv allows users to design the action space at both a macro (the ordering of passes and composition of passes in a pipeline) and a micro level (the ability to decide whether to apply individual rewrites from a pass).

## 2.3 INTERCEPT REWRITES FROM XLA PIPELINES WITH DRY-MODE

Desirably, at each optimization step, the decision-making agent is provided with the source HLO graph, the matched sub-graphs of all rewrite rules, and the target sub-graphs that will replace the matched patterns. The actual rewriting of the graph only happens after the agent has made decisions on which rewrites to apply.

However, existing XLA passes operate greedily by making an immediate replacement once a rewrite rule has its match. Hence it is non-trivial to refactor XLA into the above-described optimization iterations. Fortunately, we notice that all rewrites are carried out via only a few core APIs that modify the graph. Therefore, by intercepting those graph modification APIs, we introduce a *dry-mode* to XLA. Instead of making immediate replacements, we save necessary information of all the rewrite opportunities under dry-mode while keeping the source graph unchanged. Given all the opportunities identified from the dry-mode, the agent makes decisions on which rewrites to apply, which get executed by the HloEnv afterward. The dry-mode is general to most passes in XLA and maximizes the reuse of XLA's existing code by introducing minimal code change. Dry-mode also brings an additional positive side effect: parallel execution of passes becomes possible since the source graph is read-only under dry-mode.

## 2.4 THE ALTERNATIVE GRAPH REPRESENTATION

Existing works such as REGAL (Paliwal et al., 2019) and GO (Zhou et al., 2020a) use the source graph as the state and predict an action for each node of the graph. This state and action space work well for optimizations like device placement or execution priority. However, they are insufficient for complex graph rewrites like algebraic simplification.

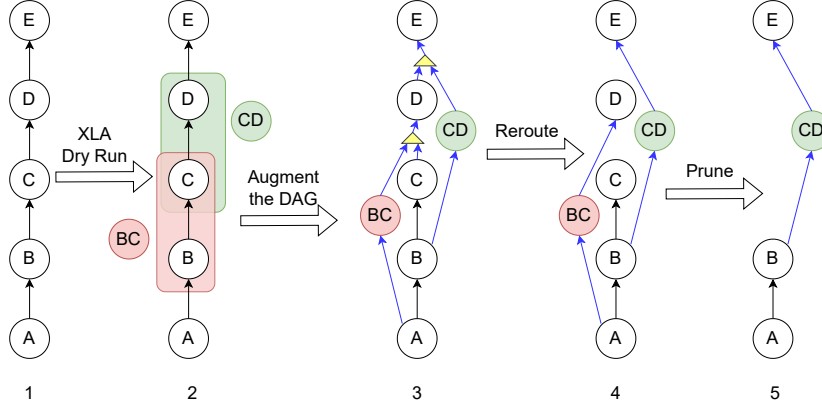

Figure 2: The alternative graph-based optimization pipeline.

To generalize to any type of graph rewrite, and to enable joint consideration of multiple rewrite rules that potentially conflict with each other, we introduce the representation shown in Fig. 2. When a source graph is processed in a pass with *dry-mode* enabled, we capture each rewrite opportunity instead of applying them. We then augment the graph with the identified rewrite opportunities using a special kAlternative instruction (yellow triangle), resulting in an *alternative graph*. This alternative graph serves as the state for the agent. After decisions are made on which input path to take for each kAlternative instruction in the reroute step (details in Section 3), HloEnv applies a pruning step to remove all unused nodes and perform additional clean-up.

**Interfering Rewrites**    Commonly, two rewrites on the same graph can interfere with each other, i.e., their matched pattern overlaps. Therefore, in existing XLA pipelines, rewrites happen sequentially in a pre-defined order to avoid race conditions. In our pipeline, two interfering rewrites are both inserted

as alternatives. This enables our agent to make a more knowledgeable decision based on all available opportunities. Two or more interfering rewrites can cause the resulting graph to violate the acyclic constraint. Hence we introduce a cycle detection function to reject all such alternatives.

**Writing New Passes in XLA**    The generality of our alternative graph representation means that we can easily enable *dry-mode* for any new pass we create, as long as it utilizes XLA's core APIs. This is important as it allows us to easily introduce modified versions of XLA passes into HloEnv, with larger action spaces for an agent or heuristic to make the rewrite decisions. See our custom *General Fusion* pass in Section 5.2 for an example.

## 2.5 CUSTOM HASH FUNCTION FOR HLO GRAPH

We require an HLO graph hash function for de-duplicating the dataset or uniquely labeling the state when performing a search over the state space. However, the existing hash implementation in XLA does not satisfy our needs. It does not fully account for the graph topology or the parameters specific to individual instruction types. This can potentially lead to a higher rate of hash collisions. At the same time, it also is affected by irrelevant information such as the ordering of the instructions/computations. Hence, we developed `HloDagHash`, a more powerful hash that captures the graph topology and all instruction parameters to allow for the unique identification of each graph. The details about its implementation can be found in Appendix B.

## 3 OPTIMIZATION STRATEGIES

Section 2.4 details how we organize alternative rewrites in the alternative graph (i.e. step 2 & 3 in Fig. 2, and the `Augment` method in Algorithm 1 below). In this section, we utilize HloEnv's ability to control individual rewrites to explore alternative optimization strategies, including heuristic-based and search-based methods. These methods select one input path for each alternative node, as represented in the `GenerateAction` method in Algorithm 1 below.

As shown in Algorithm 1, we frame the HLO graph optimization for a certain pass (identified as `pass_id`) as a Markov Decision Process (MDP). Given a certain augmented graph $\hat{G} \in \mathcal{S}$, the corresponding action space is $\mathcal{A}^{\hat{G}} = \mathcal{A}_1 \times \cdots \times \mathcal{A}_D$, where $D$ is the number of `kAlternative` nodes in $\hat{G}$.

---

**Algorithm 1** Frontend HLO Graph Optimization Pass

---

1: **function** OPTIMIZATION PASS($G^0$, `pass_id`)                                   $\triangleright G^0 = (V, E)$
2:     $\hat{G}^0 \leftarrow$ `Augment`($G^0$, `pass_id`)
3:     step $\leftarrow 0$
4:     **while** $\hat{G}^{\text{step}} \neq G^{\text{step}}$ **do**                 $\triangleright$ Loop while still having alternative nodes
5:         $a \leftarrow$ `GenerateAction`($\hat{G}^{\text{step}}$, `pass_id`)      $\triangleright$ Action space $\mathcal{A}^{G^{\text{step}}} = \mathcal{A}_1 \times \cdots \times \mathcal{A}_D$
6:         $G^{\text{step}+1} \leftarrow$ `ApplyAction`($\hat{G}^{\text{step}}$, $a$)          $\triangleright$ Step 4 & 5 in Fig. 2
7:         $\hat{G}^{\text{step}+1} \leftarrow$ `Augment`($G^{\text{step}+1}$, `pass_id`)
8:         step $\leftarrow$ step $+ 1$
9:     **end while**
10:     **return** $G^{\text{step}}$                                              $\triangleright$ The final optimized HLO graph
11: **end function**

---

## 3.1 HEURISTIC-BASED METHODS

Heuristic-based methods use human-designed rules for decisions on each alternative node in the alternative graph. We present a simplistic pick-first heuristic. This baseline heuristic always takes the first choice available on each `kAlternative` instruction which leads to graph change (i.e., first alternative after the original sub-graph). This acts as a baseline for other methods and the original XLA pipeline.

### 3.2 SEARCH-BASED METHODS

Search-based methods explore multiple actions at each graph state while backtracking is allowed and determine an optimal decision sequence starting from $G^0$ to an end graph $G^N$. We exhibit two search-based methods: beam search (BS) and factorized Monte-Carlo tree search ($f$-MCTS).

**Beam Search** For graph $G^t$ with action space $\mathcal{A}$, beam search (BS) enumerates all next graph states $\{G_i^{t+1}\}_{i=1}^{|\mathcal{A}|}$ and uses a runtime upper bound as pruning rule to discard children violating Eq. 1 in the search tree, equipping our beam search with adaptive bandwidth.

$$T(G_i^{t+1}) < \alpha \cdot T(G^t). \tag{1}$$

Our beam search uses depth-first strategy, implemented with a stack to prioritize search depth (details in Appendix C.2). We use large $\alpha$ to approximate an exhaustive search at the cost of a large search space. Therefore, beam search could only search on small graphs.

**Factorized Monte-Carlo Tree Search** To deal with graphs with arbitrary sizes, we base our search algorithm on the widely used (Silver et al., 2016; 2018; Schrittwieser et al., 2020) Monte-Carlo tree search (MCTS). However, our computation graph optimization problem poses new challenges to existing MCTS methods. First, the optimization spaces differ for each computation graph as the action space varies from search node to search node. Second, the actions are naturally factorized in our problem, while the upper confidence bounds in MCTS are usually developed for flattened actions. Third, one search node might have multiple different parent nodes. Therefore, we propose *factorized MCTS* ($f$-MCTS) to address the above challenges.

MCTS maintains statistics for visit counts and action (Q) values, representing the expected cumulative reward for taking a specific action at a state. The key idea of $f$-MCTS is to replace the joint Q value with a set of marginal Q values based on the number of `kAlterantive` nodes in the current computation graph. In this way, the space complexity for storing all Q values is linear to the number of `kAlterantive` nodes, and different numbers of factors are allowed for different graphs. Thus, the tree search can operate on dynamic action spaces. In the simulation phase, each `kAlterantive` node can make decisions independently. However, in the backup phase, related marginal statistics will be updated to fulfill the marginalization operation (details in Appendix C.3). This way, our method can address the above three difficulties and work efficiently on arbitrarily large graphs.

## 4 ANALYSIS OF XLA OPTIMIZATION PASSES

The previous section assumes a certain pass is selected and layouts the optimization loop. In this section, our experiments lead us to the design of our action space, namely, which passes are of interest.

We conduct a pass analysis on the optimization passes in XLA's frontend to determine their impact on runtime performance. Thanks to the flexible interface provided by HloEnv, we can easily reorder and disable any pass in our Python analysis script and evaluate its effect on the resulting HLO graph's runtime. Without loss of generality, we only consider optimization-focused passes (ignoring passes strictly for ensuring runtime correctness) and restrict our analysis to NVIDIA GPUs, the most commonly used backend for existing DL compilers. From this analysis, we select optimization passes with the most significant impact to explore how changes in their rewrites can potentially improve performance over XLA's heuristics (Section 3).

**Overview** There are 222 passes in total, of which 143 passes operate on the HLO graph when compiled for GPU. We ignore correctness-critical passes and select 21 runtime/memory optimization-focused passes for our analysis. For each of these passes, we utilize HloEnv to remove all instances of the pass type from the optimization pipeline and measure how this removal changes the runtime of the resulting HLO graph as compared to a fully optimized HLO graph with all optimization passes (see Table 1).

**Measurement of Performance Impact** We analyze the impact on the performance of an optimization pass/pipeline on the HLO dataset from two perspectives: the proportion of the dataset affected by that pass (% Affected HLOs), and the average change in performance as a result of that pass (runtime ratio w/ and w/o the pass). The results can be shown by a few metrics presented in Table 1:

Table 1: Analysis on selected XLA optimization passes. A higher runtime ratio indicates that the pass *improved* runtime since its absence in the optimization pipeline resulted in a higher relative runtime.

| Removed Pass/Pipeline | % Affected HLOs | | | Runtime ratio w/ and w/o the pass | |
|---|---|---|---|---|---|
| | %Changed | %Impr. (<0.96) | %Degr. (>1.06) | Avg. Ratio | Avg. Ratio (Changed) |
| ZeroSizedHloElimination | 7.46 | 0.32 | 0.46 | 1.000 | 1.004 |
| AlgebraicSimplifier | 36.85 | 0.92 | **5.91** | 1.012 | 1.033 |
| DotMerger | 5.31 | 0.03 | 0.36 | 1.001 | 1.019 |
| SortSimplifier | 5.35 | 0.05 | 0.36 | 1.001 | 1.019 |
| TupleSimplifier | 5.73 | 0.08 | 0.38 | 1.001 | 1.021 |
| WhileLoopSimplifier | 7.20 | 0.07 | 2.24 | 1.012 | 1.165 |
| HloConstantFolding | 11.24 | 0.14 | 0.63 | 1.002 | 1.018 |
| ConditionalSimplifier | 5.47 | 0.14 | 0.42 | 1.001 | 1.020 |
| TransposeFolding | 5.42 | 0.08 | 0.42 | 1.000 | 1.023 |
| AllReduceFolder | 5.46 | 0.16 | 0.43 | 1.001 | 1.024 |
| AllReduceReassociate | 5.48 | 0.13 | 0.42 | 1.001 | 1.023 |
| AllGatherBroadcastReorder | 5.54 | 0.18 | 0.46 | 1.001 | 1.025 |
| CudnnVectorizeConvolutions | 5.46 | 0.15 | 0.44 | 1.001 | 1.023 |
| CublasPadForGemms Pipeline | 5.52 | 0.18 | 0.46 | 1.001 | 1.023 |
| GpuTreeReductionRewriter | 5.71 | 0.21 | 0.52 | 1.001 | 1.024 |
| GemmRewriter | 8.56 | 2.03 | 1.78 | 1.047 | 1.552 |
| GemmBroadcastFoldingRewriter | 5.54 | 0.20 | 0.44 | 1.001 | 1.024 |
| Fusion Pipeline | 49.22 | 0.25 | **44.61** | 1.579 | 2.178 |
| AllGatherCombiner | 5.49 | 0.28 | 0.46 | 1.001 | 1.019 |
| AllReduceCombiner | 5.57 | 0.26 | 0.53 | 1.001 | 1.023 |
| ReduceScatterCombiner | 5.59 | 0.27 | 0.51 | 1.001 | 1.023 |

- the percentage of graphs that have been transformed by the pass (%Changed) as determined by comparing their `HloDagHash`;
- the percentage of graphs that have improved/degraded performance (%Impr./%Degr.);
- the average improvement in runtime they result in across all graphs (Avg. Ratio);
- the average improvement in performance specifically for the graphs that change when the pass is removed, i.e., graphs which the pass affects on (%Avg. Ratio - Changed).

Some passes have a significant impact on performance on the graphs that they affect but only affect a minimal number of graphs (e.g., WhileLoopSimplifier). Hence these passes have a lower average difference in the performance change. Due to runtime noise, we evaluate a graph as having improved performance when the relative runtime ratio against XLA (i.e., the runtime of that graph divided by the runtime of the fully XLA optimized graph) is less than 0.94, and degraded performance when it is above 1.06 (see Appendix D.3 for more details on how we set these limits).

**Passes/Pipelines of Significance**    There are two passes/pipelines which have the most significant impact on the HLO graphs on which they operate. Hence we choose to focus our experiments on these two passes/pipelines. These are the *AlgebraicSimplifier* pass and the *Fusion* pipeline (consisting of a variety of passes related to instruction fusion). Of most significance is the Fusion pipeline, which affects the most significant percentage of HLO graphs and results in the largest performance improvement. AlgebraicSimplifier similarly affects a large percentage of HLO graphs but results in a more negligible general performance improvement (see Table 1).

**Insights**    Results from Table 1 show that these optimization passes do *not* always result in a runtime improvement. For example, removing the GemmRewriter pass results in 2.1% of the HLO graphs showing more than 6% of runtime improvement. This applies even for a trivial pass like HloConstantFolding, which seems like it should always be applied. Our pass analysis found cases where removal of the HloConstantFolding pass resulted in a final graph that ran approximately two times faster (see Fig. 6 in Appendix E). This demonstrates that there is much room for further optimization in many of the passes and pipelines, even at the macro level of deciding whether to run them on a given HLO graph.

## 5 EXPERIMENTS

### 5.1 DATASETS

HLO graphs in our dataset are of high variance in terms of the number of instructions they contain. To better measure the performance of various proposed methods, we developed a tool that traverses a full HLO graph and extracts random sub-graphs that contain a given range of instruction numbers. Applying this tool to the original HLO text files, we generate a sub-dataset with different ranges of instruction numbers: 10 to 20 (94332 sub-graphs) and 20 to 40 (3118 sub-graphs). We refer these 2 sub-datasets as *inst-10-20* and *inst-20-40*. For each of the passes/pipelines analyzed (Algebraic Simplification and Fusion), we further filter these sub-datasets to obtain an experiment set for each of these two passes. For each pass, we remove sub-graphs whose runtime correctness is affected by these passes. This allows us to focus fully on the optimization aspects of a pass. We also filter out sub-graphs for which our search-heuristic optimization cannot conclude in a reasonable amount of time. More details can be found in Appendix A.3.

### 5.2 PASS SELECTION AND MODIFICATION

We select the Fusion pipeline of passes and the Algebraic Simplification pass to evaluate the performance of our alternative optimization strategies. These passes have the most significant impact on performance, as shown in Table 1.

**Algebraic Simplification Pass** There are five separate Algebraic Simplification passes at different locations in the optimization pipeline. For our experiments, we selected the third Algebraic Simplification pass in the entire pipeline for optimization and disabled the other four Algebraic Simplification passes. This pass was selected for two reasons: 1) It can be isolated from the passes before and after. In contrast, the first Algebraic Simplification pass is located in a smaller pipeline that is run multiple times in a loop and has potential inter-dependency with these other passes. Selecting this pass for optimization results in a higher percentage of correctness issues; 2) The third Algebraic Simplification pass is run to convergence, i.e., it runs multiple times until it no longer modifies the HLO graph. This gives us a larger space for optimization over multiple runs. To obtain a fair comparison, the same four passes were disabled in the XLA pipeline. The resulting pipeline was used to obtain the reference results.

**Fusion Pipeline** XLA's Fusion Pipeline consists of various passes (e.g., MultiOutputFusion, HorizontalFusion, etc.) that fuse different instructions and computations patterns. These passes are sequentially run in a fixed order. Additionally, they contain many hand-written heuristics that determine whether a fusion should occur. In our alternative graph-based representation, however, the philosophy is to keep both alternatives and delay the heuristics to the routing step. Therefore, we remove heuristic decisions from XLA's existing pipeline but keep only the rewrite rules. As a testimony, we introduce a *General Fusion* pass that is heuristics-free to replace the existing fusion pipeline. General Fusion has much shorter lines of code than the original fusion passes and provides a much larger search space.

### 5.3 EVALUATION AND METRICS

We note $T(G)$ the runtime, given a graph $G$. Based on our experiment results on profiling of the runtime noise (Appendix D.3), we define $T(G) = \min(G.\texttt{evaluate(10).async\_timing})$. For both pass/pipeline, we compute the runtime ratio of the final optimized graph w.r.t. XLA $\rho = T(G_{\text{method}})/T(G_{\text{XLA}})$ and report its avg, max and min across the dataset. We also report the proportion of the graphs with performance compared against XLA with criteria. We statistically set $\rho < 0.94$ (resp. $\rho > 1.06$) as criteria for faster (resp. slower) than XLA to avoid false positives caused by noise (Appendix D.3 and Fig. 5). We identify $G_{\text{method}} = G_{\text{XLA}}$ using our custom $\texttt{HloDagHash}$ (Section 2.5) and report the proportion of graphs that end with an equal hash. To further reduce the impact of noise on the maximum and minimum runtime ratios, we retest the graphs with the largest and smallest relative runtime ratios to remove outliers.

Table 2: Results on all sub-datasets of 2 passes. $f$-MCTS is *factorized MCTS* with uniform prior.

| Pass | Dataset | Method | Runtime ratio w.r.t. XLA | | | % of graphs of runtime ratio w.r.t. XLA | | Identical to XLA (equal hash) |
|---|---|---|---|---|---|---|---|---|
| | | | Avg. | Max. | Min. | Faster (< 0.94) | Slower (> 1.06) | |
| *Alg-Simp* | *inst-10-20* | pick-first | 1.000 | 1.158 | 0.740 | 0.1 % | 0.04 % | 96.2 % |
| | | BS | 0.981 | 1.442 | 0.383 | 11.2 % | 1.0 % | 34.9 % |
| | *inst-20-40* | pick-first | 1.000 | 1.197 | 0.883 | 0.19 % | 0.25 % | 92.2 % |
| | | $f$-MCTS | 0.995 | 1.449 | 0.570 | 5.3 % | 1.2 % | 55.5 % |
| *General Fusion* | *inst-10-20* | pick-first | 0.997 | 1.709 | 0.679 | 3.1 % | 0.3 % | 26.4 % |
| | | BS | 0.986 | 1.252 | 0.430 | 5.2 % | 0.3 % | 11.7 % |
| | *inst-20-40* | pick-first | 0.989 | 3.313 | 0.738 | 13.4 % | 1.9 % | 11.6 % |
| | | $f$-MCTS | 0.992 | 1.755 | 0.349 | 14.5 % | 7.7 % | 4.9 % |

## 5.4 RESULTS AND ANALYSIS

**Finding exemplars of performance gains over XLA**  In cases where the alternative optimization strategy generated a graph with faster performance, we performed additional evaluations, using $T(G) = \min(G.\texttt{evaluate(100).async\_timing})$, to confirm that their performance gains were not due to noise. These cases are shown below as proof that an alternative optimization strategy can achieve significant performance gains over XLA.

### 5.4.1 HEURISTIC-BASED METHODS

On both *inst-10-20* and *inst-20-40* (where graphs are not exhaustively searchable), a pick-first heuristic serves as a strong baseline.

**Algebraic Simplification**  Table 5.4 shows that the pick-first heuristic performs on par with XLA in the Algebraic Simplification pass, with more than 90 % of the final pick-first optimized graphs being identical to XLA's. This is expected as in the Algebraic Simplification case, pick-first after dry-run has a high probability of recovering the behavior of original XLA.

**General Fusion**  For the general fusion pass, the pick-first heuristic performs even better than XLA (i.e. with an average runtime ratio w.r.t. XLA of less than 1) for both *inst-10-20* and *inst-20-40*. In this case, we are comparing XLA's fusion pipeline, which contains a variety of heuristics used to evaluate if fusion should be performed, with our modified general fusion pass, which removes most of these heuristics. Hence, the good performance of the pick-first heuristic on both sets shows that a large portion of graphs can be optimized well despite ignoring XLA's hand-designed heuristics and simply fusing everything.

### 5.4.2 SEARCH-BASED METHODS

In general, using search-based methods resulted in a faster average runtime over XLA (as well as heuristic-based methods) for both the *inst-10-20* and *inst-20-40* sets for both the Fusion and Algebraic Simplification passes. We remove pruning from beam search on *inst-10-20* for exhaustive search.

**Algebraic Simplification**  On *inst-10-20*, using search-based methods to optimize the HLO graphs showed general reductions in their evaluation timings, with superior performance compared to the pick-first heuristic. On average, BS-optimized graphs were 1.9% faster than the equivalent XLA-optimized graphs, with the most optimized graphs performing up to 161.1% better (Fig. 7). On *inst-20-40*, performance was slightly worse than on *inst-10-20*, with $f$-MCTS being on average 0.5 % faster than XLA. Nevertheless, the most optimized graphs were still significantly faster than XLA and ran up to 75.4% faster (Fig. 8).

We examine the differences between the best-performing search-optimized and XLA-optimized HLO Graphs to understand the source of these performance gains. In these cases, not performing certain

Algebraic Simplification rewrites either directly improve performance, or allow for subsequent passes to make better optimizations (e.g., by allowing for a later Fusion pass to fuse more instructions into a single kernel, or allowing a different cuDNN call to be used). Exemplars of these cases can be found in Appendix F.1.

**General Fusion** On *inst-10-20*, using search-based methods showed general reductions in their evaluation timings. On average, best-first search optimized graphs were 1.42% faster than the equivalent XLA-optimized graphs, with the best graphs performing 132.5% faster (Fig. 13). On *inst-20-40*, $f$-MCTS has approximately equal performance to the pick-first heuristic. This is despite the best performing $f$-MCTS graph running 186% faster than the XLA optimized equivalent (Fig. 14). One reason for this could be that greedily performing as much fusion as possible is optimal in most cases.

As we did with the Algebraic Simplification results, we examine the best-performing search-optimized graphs and compare them to XLA-optimized graphs to understand why they performed better. These cases can be split into two general types: 1) Trivial fusion cases, where the speed up happens due to the instructions being fused into a smaller number of kernels and 2) Non-trivial fusion cases where the number of kernels is the same or fewer, but changes to the topology of the resulting graph result in a faster runtime. Exemplars of both these types can be found in Appendix F.2.

## 6 RELATED WORKS

The efficient deployment of various DL models on diverse DL hardware is a challenging task (Jia et al., 2019a; Jouppi et al., 2017; Chen et al., 2014). It has driven development and research on DL compilers such as TVM (Chen et al., 2018), XLA (Leary & Wang, 2017), DLVM (Wei et al., 2017), nGraph (Cyphers et al., 2018), and Glow (Rotem et al., 2018). Several third-party systems and methods have also been proposed to enlarge the search space (Jia et al., 2019b; Wang et al., 2021), include better optimization strategies (Looks et al., 2017; Liu et al., 2019; Zheng et al., 2020), or providing new high-level IR (Roesch et al., 2019). More recently, people have started exploring learning-based methods for DL compiler frontend optimization (Paliwal et al., 2020; Zhou et al., 2020b). CompilerGym (Cummins et al., 2022) and MLGO (Trofin et al., 2021) are two works related to ours in the traditional compiler optimization field. They provide Python environments for users to interact with a selected subset of LLVM optimization passes. TensNet (Zheng et al., 2021) is a dataset for tensor programs that target lower-level optimization for tensor compilers.

## 7 DISCUSSIONS AND CONCLUSIONS

We develop HloEnv, the first (to our knowledge) general interface that inter-operates with a production-quality DL compiler. HloEnv provides a common representation for graph rewrites. We utilize this common representation to test alternative optimization strategies for specific, high-impact passes on a broad range of HLO graphs. We show that these alternative strategies can achieve on-par or better results than the DL compiler's native optimization strategy. Allowing the user to control the action space of the graph rewrites provides a more generic and flexible (as a result, more challenging) setup. This action space can be restricted (e.g., if users only select certain passes of interest) or, in principle, infinite (by defining and introducing new passes with novel graph rewrite rules). To supplement research with HloEnv, we also generate a large-scale HLO graph dataset to act as an ideal testbed for computation graph optimization.

We hope that HloEnv and this HLO graph dataset provide valuable tools to the community to spur progress in developing DL compiler systems. Building on the baseline presented in this paper, we believe that a well-designed action space (system research) and a well-trained agent (ML research) are both essential for this purpose. More specifically, we hope for HloEnv to enable DL compiler development in the following directions: 1) open up opportunities for more types of decision-making agents that improve native optimization passes for existing DL compiler systems such as XLA; 2) less effort to introduce new passes as the heuristics will be replaced by the optimization agent, e.g., General Fusion, and 3) lead to the development of learning-based policies to generate optimization strategies that generalize to new DL models running on new DL hardware.

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

# A  DATASET INFORMATION

The goal of our HLO dataset is twofold. First, we want to present a large-scale dataset for training different computation graph optimization strategies. Second, we want the dataset to serve as an ideal test-bed against which people could measure the performance of arbitrary computation graph optimization strategies.

## A.1  DATASET COLLECTION

We manually select a list of JAX implemented repositories from GitHub, and harvest the HLO text files by setting the XLA_DUMP_TO flag while running the model. In this way, we dump all the unoptimized HLO graphs generated during JAX's Just-In-Time (JIT) compilation process. We then remove duplicate HLO text files by comparing hash using our `HloDAGHash` implementation (see Appendix B for more details), and filter the resulting files to remove the very small ones which have minimal opportunities for optimization. After the above steps, we can guarantee three properties of HLO graphs in our dataset: 1) They all come from real-world deep learning models; 2) They all have different DAG and tensor shapes, and 3) They all provide at least some space for optimization opportunities. In total, we build a dataset containing 40,711 HLO graphs from deep learning models defined in 26 distinguished GitHub repositories.

## A.2  DATASET OVERVIEW AND ANALYSIS

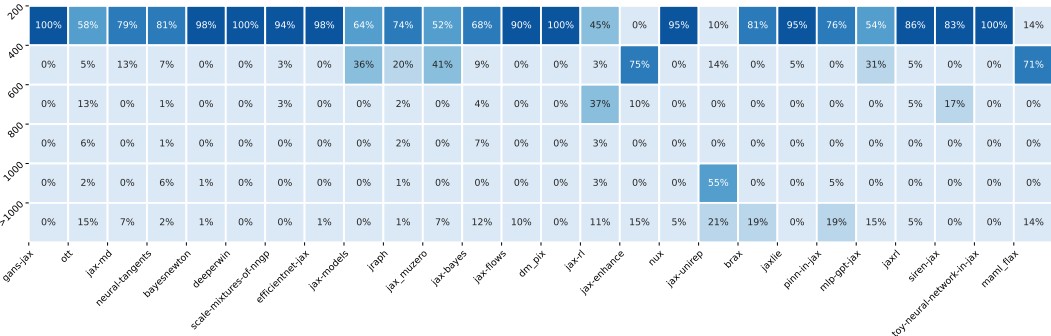

Figure 3: Distribution of the sizes of HLO graphs

Our dataset covers a broad range of network architectures, centering around modern, real-world models in various domains. The GitHub repositories we pick include Vision Transformer (ViT) (Dosovitskiy et al., 2021), Neural-Tangent (Novak et al., 2020), MuZero (Schrittwieser et al., 2020), and more. Figure 3 shows that most HLO graphs generated during JAX's JIT compilation contain less than 1000 instructions. More than half of the repository set majorly contains a dataset with less than 200 instructions. We also show a breakdown of top HLO opcodes that appear in our dataset in Figure 4.

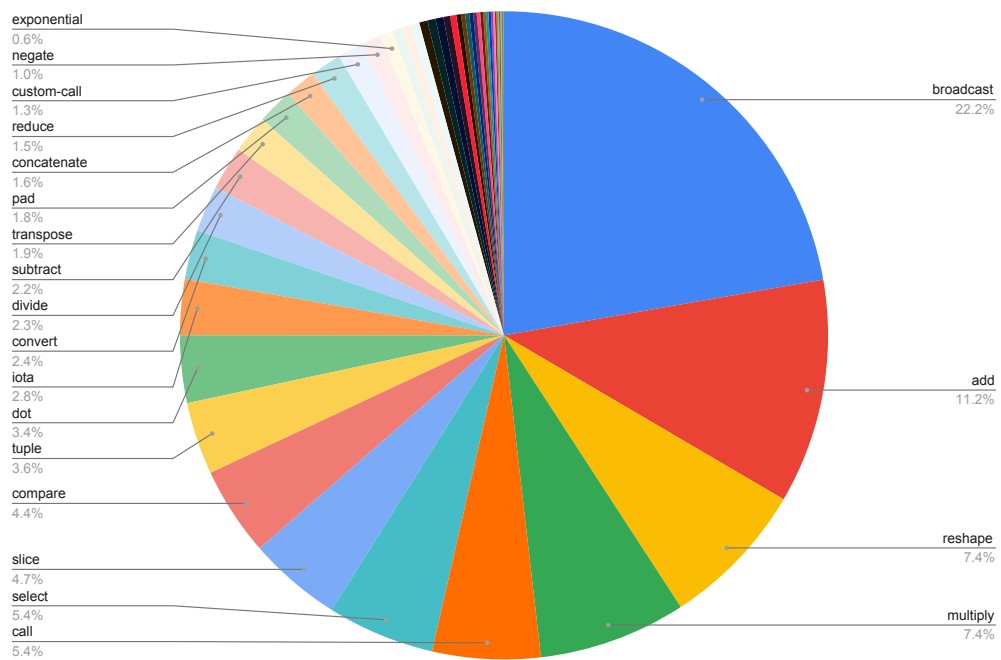

Figure 4: Distribution of the Ops in our HLO dataset.

## A.3 SUB-DATASET FILTERING

**Algebraic Simplification** The Algebraic Simplification pass is used for both optimization and correctness purposes. Hence it is possible that modifications to the rewrites applied by this pass result in a graph that cannot be correctly evaluated on the backend. Of the 94332 *inst-10-20* sub-graphs generated, we filtered out 34000 graphs for which the correctness of the graph was not dependent on any graph rewrites performed by the Algebraic Simplification pass. Filtering *inst-20-40* results in 1619 graphs.

**General Fusion** Our implementation of a General Fusion pass allows almost any two instructions/computations to be fused. Additionally, fusion can result in the cloning of computation. As a result, the action space for General Fusion is significantly larger than for the Algebraic Simplification pass. To ensure our search algorithms can finish in a reasonable time, we empirically filter graphs from the sub-datasets with action space less than 10,000 in size after the first optimization pass. Note that the number of instructions in a graph can significantly increase over multiple rewrite operations during a search. This filtering leads to *inst-10-20* containing 28,507 graphs and *inst-20-40* containing 2791 graphs.

## B DETAILS ON THE IMPLEMENTATION OF THE `HloDAGHash` FUNCTION

The XLA HLO graph hash implementation is lacking in two ways which increase the number of hash collisions: 1) It simply hashes the instructions in the HLO graph in post-order, and *does not* recursively consider the structure and connections of each HLO instruction and computation in the HLO graph; 2) Instruction specific parameters (e.g. the size and stride of an HLO Convolution instruction) are not considered in the hash of each instruction as well.

Our custom `HloDAGHash` function builds upon XLA's hash implementation, but is designed to be a more powerful hash that additionally accounts for graph topology and the parameters unique to each

Table 3: Commit hashes and GitHub URLs of open-source repositories we used to generate our dataset.

| GitHub Repo Name | Commit Hash | URL |
|---|---|---|
| BayesNewton | e3a7251 | AaltoML/BayesNewton |
| GANs-JAX | 099111f | lweitkamp/GANs-JAX |
| antibiotic-resistance | 7634f9c | magi-1/antibiotic-resistance |
| bayeSDE | 43511d2 | xwinxu/bayeSDE |
| brax | 730e05d | google/brax |
| continuation-jax | c145260 | harsh306/continuation-jax |
| cwvae-jax | d724112 | juliuskunze/cwvae-jax |
| deeperwin | 5c8d497 | mdsunivie/deeperwin |
| deepmind-research | 1642ae3 | deepmind/deepmind-research |
| dks | a92c184 | deepmind/dks |
| dm_pix | 6accc96 | deepmind/dm_pix |
| efax | 61ad838 | NeilGirdhar/efax |
| efficientnet-jax | a65811f | rwightman/efficientnet-jax |
| flaxvision | d5b7e6c | rolandgvc/flaxvision |
| jax-bayes | b91432c | jamesvuc/jax-bayes |
| jax-enhance | 3a3dd40 | isaaccorley/jax-enhance |
| jax-flows | 26dce81 | ChrisWaites/jax-flows |
| jax-gat | 4d48e2b | gcucurull/jax-gat |
| jax-md | 2b8754a | google/jax-md |
| jax-models | ae57505 | DarshanDeshpande/jax-models |
| jax-rl | 820cb5d | henry-prior/jax-rl |
| jax-unirep | b8048db | ElArkk/jax-unirep |
| jax_muzero | b8ab362 | Hwhitetooth/jax_muzero |
| jax_verify | e02aa65 | deepmind/jax_verify |
| jaxfg | 893c347 | brentyi/jaxfg |
| jaxlie | 65d6351 | brentyi/jaxlie |
| jaxrl | 1a86300 | ikostrikov/jaxrl |
| jraph | 36071d5 | deepmind/jraph |
| maml_flax | a4a8819 | gcucurull/maml_flax |
| medclip | ff9e5f6 | Kaushalya/medclip |
| metax | 12f2478 | tristandeleu/metax |
| mlp-gpt-jax | 571ccf0 | lucidrains/mlp-gpt-jax |
| neural-tangents | 5bb274c | google/neural-tangents |
| ott | f4dafa8 | ott-jax/ott |
| revisiting_rainbow | 7cd2bc6 | JohanSamir/revisiting_rainbow |
| scenic | 978fea0 | google-research/scenic |
| siren-jax | 0806e61 | KeunwooPark/siren-jax |
| tree-math | 46214b5 | google/tree-math |
| Scale-Mixtures-of-NNGP | 4412de7 | yuneg11/Scale-Mixtures-of-NNGP |
| Toy-neural-network-in-jax | 3a1d5d8 | rasutt/Toy-neural-network-in-jax |
| NuX | bd996dd | Information-Fusion-Lab-Umass/NuX |
| PINN-JAX | 561a8d9 | ASEM000/Physics-informed-neural-network-in-JAX |
| jax_cosmo | 52ca009 | DifferentiableUniverse Initiative/jax_cosmo |

instruction. This reduces the chance of a hash collision when determining if a graph has been seen before, or is identical to another graph.

**Implementation** The `HloDAGHash` algorithm walks the HLO graph, starting from the root instruction, in Depth-First Traversal order. At each instruction, we take the original XLA hash of that instruction and additionally hash it with two things. The first is the `HloDAGHash` of each operand of that instruction (unlike the XLA hash which hashes the shape of each operand of that instruction). In this way, we can ensure that two HLO graphs with the same post-order of instructions, but different structures, will not have the same hash. The second is that the XLA hash of each HLO instruction is further hashed with the attributes specific to that instruction opcode (e.g. the slice starts, limits, and strides of an HLO Slice instruction). This further decreases the chance of a hash collision between two differing HLO graphs.

## C  DETAILS ON ALTERNATIVE OPTIMIZATION STRATEGIES

### C.1  NOTATIONS

We note $G = (V, E)$ the HLO computation graph parsed from HLO text file by our utility; $\hat{G} = (\hat{V}, \hat{E})$ the alternative graph augmented from $G$, $\hat{V} = V \cup \{d\}_1^D$, with $D$ kAlternative nodes. In reinforcement learning setup, $\hat{G}$ corresponds to the state. The action space given $\hat{G}$ (or $G$) is denoted as $\mathcal{A}^G = \mathcal{A}_1 \times \cdots \times \mathcal{A}_D$. For all $d = 1, \ldots, D$, $\mathcal{A}_d = \{k \to d \wedge k \in V\}$ and $|\mathcal{A}_d| = \texttt{in\_degree}(d)$. We note an action $a = (a_d)_{d=1}^D \in \mathcal{A}^G$. We omit the superscript for $\mathcal{A}^G$ and note $\mathcal{A}$ when there is no ambiguity.

In our sequential decision problem, we use superscript for $G$ and $a$ to denote the search steps: Starting from a graph $G^0$, we apply an action $a^0$ to produce the next graph $G^1$; we repeat the above for $N$ steps to get the final graph $G^N$. We note $T$ as the function for calculating the running time of graph $G$ described in D.3, specifically, Eq. 7.

### C.2  BEAM SEARCH

We maintain a stack $S$ with no size limit and a global minimum runtime $T_{\min}$ during the search. $S$ has one element $G^0$ initially. At every search step $t$, we pop from $S$ the graph $G^t$; we apply all possible actions $a \in \mathcal{A}$ to its alternative graph $\hat{G}^t$ to obtain new graphs $\{G_i^{t+1}\}_{i=1}^{|\mathcal{A}|}$; we evaluate the runtime of each new graph and only push it back into $S$ with criteria

$$T(G_i^{t+1}) < \alpha \cdot T(G^t). \tag{2}$$

$\alpha$ controls the pruning when runtime degradation happens. The number of new graphs being pushed back is capped by a pre-fixed expand budget. The search ends when the stack $S$ is empty or a global timeout is triggered. Beam search becomes exhaustive when we push all new graphs into the stack (equivalent to setting $\alpha = +\infty$ with no expand budget) at every step.

When the search finishes, we extract the graph with runtime $T_{\min}$ and its trajectory starting from $G^0$.

Beam search can achieve optimization at the cost of large search space as the number of kAlternative nodes increases (e.g. for a graph with 10 kAlternative nodes with each node 2 choices, the search space is of size $2^{10}$). Therefore, an exhaustive search is only feasible on graphs with a considerably small number of alternatives.

### C.3  FACTORIZED MCTS

Factorized MCTS ($f$-MCTS) maintains a search tree to decide which action to take to transit from $G^t$ to $G^{t+1}$. For a trajectory $(G^0, \ldots, G^N)$, $f$-MCTS maintains $N$ search trees, each with root node $G^0, \ldots, G^{N-1}$. Without loss of generality, we present the algorithm for one search tree with root node $G^0$.

We note $(G, a)$ the state-action pair and $G'$ the graph obtained after applying $a$ on $G$. We define the reward function as follows:

$$R(G, a, G') = T(G) - T(G'). \tag{3}$$

The action value function for $G$ and $a = (a_1, \ldots, a_D)$ is represented by $Q(G, a_1, \ldots, a_D)$, which grows exponentially as the number of alternative nodes increases. To deal with the joint action space $\mathcal{A}$, we propose to replace the joint Q with a set of marginal Q value function $Q_d(G, a_d) = \mathbb{E}_{a_i, i \neq d}[Q(G, a_1, \ldots, a_D)], d = 1, \ldots, D$. Each $Q_d(G, a_d)$ represents the expected value if only one action $a_d$ for $d$-th alternative node is taken. In this way, we can select actions for different alternative nodes independently.

During the search procedure, we associate each search node with a computation graph state $G$. Each search node maintains a set of statistics $\{T(G), \{N_d(G, a_d), Q_d(G, a_d), P_d(G, a_d)\}_{d=1,\ldots,D}\}$ representing the running time $T$, marginal visit counts $N$, marginal action value $Q$ and factorized policy $P$ for each of the alternative vertices. For each action $a$ there is an edge $(G, a, G')$ storing the transition information and the corresponding reward $R$. The search repeats the following three stages for a given number of budgets.

**Selection:** We use superscript $k$ to denote the search depth in the tree. The root node is thus given by $G^0$. All simulations start from the same root graph state $G^0$ and finish when a leaf graph $G^\ell$ is achieved or a cycle is formed. For each time-step $k$ along the search path, a joint action $a^k$ is obtained by selecting each $a_d^k$ according to the upper confidence bound (UCB) score described below:

$$a_d^k = \arg\max_{a_d} \left[ Q_d(G, a_d) + P_d(G, a_d) \cdot \frac{\sqrt{\sum_{b_d} N_d(G, b_d)}}{1 + N_d(G, a_d)} \left( c_1 + \log\left( \frac{\sum_{b_d} N(G, b_d) + c_2 + 1}{c_2} \right) \right) \right]. \tag{4}$$

$P_d$ is a prior policy while $Q_d$ accumulates knowledge from simulations. $c_1$ and $c_2$ are two hyperparameters to trade off the relative importance of $P_d$ and $Q_d$. At the beginning of a search, UCB relies more on the prior policy but gradually moves its attention to value statistics. In our experiments, we choose $c_1 = 1.25$ and $c_2 = 19652$ following AlphaGo (Silver et al., 2016).

**Expansion:** Expansion happens when a computation graph is visited for the first time in the search tree, *i.e.*, when a simulation terminates. Consider a terminal transition $(G^{\ell-1}, a^{\ell-1}, G^\ell)$, a new node representing $G^\ell$ will be created and added to the search tree. Once prior policies $\{p_d^\ell\}_{d=1}^D$ for kAlternative nodes $\{d\}_1^D$ and a value function $v_\theta(G)$ to obtain the value $v^\ell$ are given. The node statistics will be initialized to $\{N_d(G^\ell, a_d) = 0, Q_d(G^\ell, a_d) = 0, P_d(G^\ell, a_d) = p_d^\ell\}_{d=1}^D$. The running time is set to $T^\ell = T(G^\ell)$. The reward for the current transition is also initialized by $R(G^{\ell-1}, a^{\ell-1}, G^\ell) = T^{\ell-1} - T^\ell$. Note that, for the expansion of the root note representing $G^0$, there will not be a reward as transitions exist.

**Backup:** Each simulation generates a search path $\{G^0, G^1, \ldots, G^\ell\}$. The statistics of nodes/graphs along this path need to be updated in reverse order. Let $r^t$ denote the reward for transition $(G^{t-1}, a^{t-1}, G^t)$, and $\gamma$ be the discounting factor. The $(\ell - k)$-step return estimation at $k$-th step is given by

$$G^k = \sum_{\tau=1}^{\ell-1-k} \gamma^\tau r^{k+1+\tau} + \gamma^{\ell-k} v^\ell, \tag{5}$$

where $v^\ell$ is the value for $G^\ell$. For $k = \ell, \ldots, 1, 0$ we update the marginal statistics for each $(G^t, a_d^t)_{d=1,\ldots,D}$ as follows:

$$Q_d(G^k, a_d^k) := \frac{N_d(G^k, a_d^k) \cdot Q_d(G^k, a_d^k) + G^k}{N_d(G^k, a_d^k)}, \quad \forall d = 1, \ldots, D;$$

$$N_d(G^k, a_d^k) := N_d(G^k, a_d^k) + 1, \quad \forall d = 1, \ldots, D. \tag{6}$$

However, the reward and value might have an arbitrary scale in our setting. We propose to normalize the Q values such that $Q \in [0, 1]$ to get a stable calculation of the UCB score. To this end, we keep track of the minimum ($Q_{\min}$) and maximum ($Q_{\max}$) values observed in the search tree. A normalized Q value is thus obtained by $\bar{Q} = \frac{Q - Q_{\min}}{Q_{\max} - Q_{\min}}$. When we calculate the UCB score in Eq. 4, we are actually using normalized Q instead of un-normalized ones in Eq. 6.

# D EXPERIMENTS

## D.1 HARDWARE AND SOFTWARE ENVIRONMENT

We empirically found that competing processes running on the same machine is a major source of noise in the runtime evaluation of a graph. To get the best estimation of runtime in a real-world environment, we directly evaluate the runtime of an HLO graph on a clean bare-metal GPU node with minimal other processes running. The GPU node has two AMD EPYC 7352 24-Core processors (with hyper-threading 96 cores), 512GB of main memory, and eight 40GB memory NVIDIA A100 GPUs. All tests run on Ubuntu 20.04 with CUDA 11.2, cuDNN 8.1.1, and TensorFlow 2.9.1.

## D.2 XLA VERSION

HloEnv, along with all our experiments presented in this paper, was developed from the following version of XLA (`https://github.com/tensorflow/tensorflow/commit/0bd7a41db27060eaae55da4c4572cafba29c6690`).

## D.3 PROFILING AN HLO GRAPH

To evaluate the effectiveness of any given optimization strategy, it is critical to get an accurate runtime oracle $\Omega$ : `Optimized_HLO_Module` $\rightarrow$ `Runtime`. To approximate oracle $\Omega$, researchers in the community either build a cost model or directly evaluate the runtime. The cost model is either learning-based (i.e. trained from a supervised dataset) (Baghdadi et al., 2021) or rule-based. The latter requires a large amount of engineering effort as it needs to predict the runtime without evaluation, e.g. Grappler (Larsen & Shpeisman, 2019) for Tensorflow Graph. On the other hand, although direct runtime evaluation often suffers from real-world noises, given an environment with sufficient computing resources where noise can be controlled, it provides a way to do an accurate evaluation with minimum cost. In this paper, we use the direct runtime evaluation.

To profile the runtime of an HLO graph we need to obtain both the executable and parameters. We obtain the executable by calling the standard compiler provided by XLA while setting `run_backend_only` to prevent the re-invocation of HLO passes. For parameters, we randomly generate $\mathcal{N}(0, 1)$ for floating-point parameters and fill const values for other types. A fixed random seed is used to keep the parameters consistent across the optimization process so that we can verify the correctness of optimizations.

**Reducing timing noise** There is random variation in the evaluation timing of an HLO graph. Additionally, when an executable runs multiple times, the initial run is consistently much slower than subsequent runs of that executable. To reduce this noise in the evaluation timing, we evaluate the executable multiple times. The first three runs are treated as warm-up runs and are ignored, and the executable is then evaluated at least 10 additional times. Experiments showed that running the evaluation more than 10 times did not significantly reduce the variance in the final determined runtime. We then take the minimum of the timing results across all runs. We take the minimum of the results instead of the average due to the half-normal distribution of the timing.

Additionally, we obtain three different measurements of the evaluation timing, with each being progressively more fine-grained:

- **full execution timing:** The time measured from the moment the evaluation begins till when it concludes;
- **asynchronous evaluation timing:** The time taken from the asynchronous dispatch of the computation to the moment it returns;
- **compute timing:** The time spent in nanoseconds for the execution, without accounting for data transfer.

Experiments showed that the full execution timing measurement resulted in more evaluation timing noise, while the compute timing measurement was too fine-grained and missed out on some of the performance improvements as a result of memory-related optimizations. Hence, asynchronous evaluation timing is utilized as the main timing metric for our experiments.

Thus, the formula for obtaining the runtime formally reads:

$$T(G) = \min(G.\texttt{evaluate(10).async\_timing}). \tag{7}$$

Additionally, it was determined that noise was higher when both GPUs coupled to a single NUMA node were utilized. Hence when obtaining our experimental results, we ensured that only a single GPU in each NUMA node was utilized (four out of eight GPUs on the bare-metal system total).

**Profiling effects timing noise on evaluation of relative graph performance**   In our experiments, we have to frequently evaluate the relative performance of two HLO graphs, for instance in comparing whether the heuristic-based optimized graph performs better than XLA (Section 5.3), or the relative change in performance when a particular XLA optimization pass is removed from the full optimization pipeline as seen in Table 1.

To determine what relative ratio can be used to determine with confidence that one HLO graph has faster run-time than another HLO graph, we profile the expected noise seen when evaluating the relative run-time ratio of two HLO graphs under the same conditions as our experiments (i.e. only single GPU utilized per NUMA node). This is done by evaluating the same HLO graph twice and taking the ratio of the first runtime divided by the second runtime. This is repeated 500,000 times to obtain a distribution of the expected range in runtime ratios for a given HLO graph. We perform this profiling on 10 different graphs, spanning the range of runtimes seen in our HLO dataset (approximately 25000 ns to 1000000 ns)

From this distribution, we determine upper and lower bounds for the ratios, above/below which we can say with reasonable confidence that a degradation/improvement in run-time is not due to noise. This is evaluated by determining the ratios above and below which 99.9 of the data points lie. From our results, we can see that any run-time ratio < 0.94 and above 1.06 likely represents an actual change in performance (Fig. 5).

**Non-empirical evaluation using HLO graph Cost Analysis**   The impact of an optimization pass on an HLO graph can also be estimated by performing an HLO Cost Analysis on the resulting HLO graph, and seeing how the module changes in the metrics of 1. number of FLOPs, 2. number of Transcendentals, and 3. Bytes accessed.

### D.4   MODEL ARCHITECTURE AND RUNNING TIME

**Beam Search**   The pruning factor $\alpha$ in Eq. 1 is set to 20 to cover most cases.

**Factorized MCTS**   The simulation budget is set to 400. We decay to half (capped by 50) each step after an action is taken, and produce a new graph along the decision sequence. The budget decay is based on the observation of the size shrink of action space after the first several steps. The value function is given by the Monte-Carlo evaluation. We launch 5 rollouts of length 10 based on uniform sampling. We report the running time of $f$-MCTS with a uniform prior for reference only, as most computation is done on the CPU side while only the environment and Monte-Carlo evaluation require a GPU. $f$-MCTS with uniform prior takes 50 A100 days on *inst-20-40*.

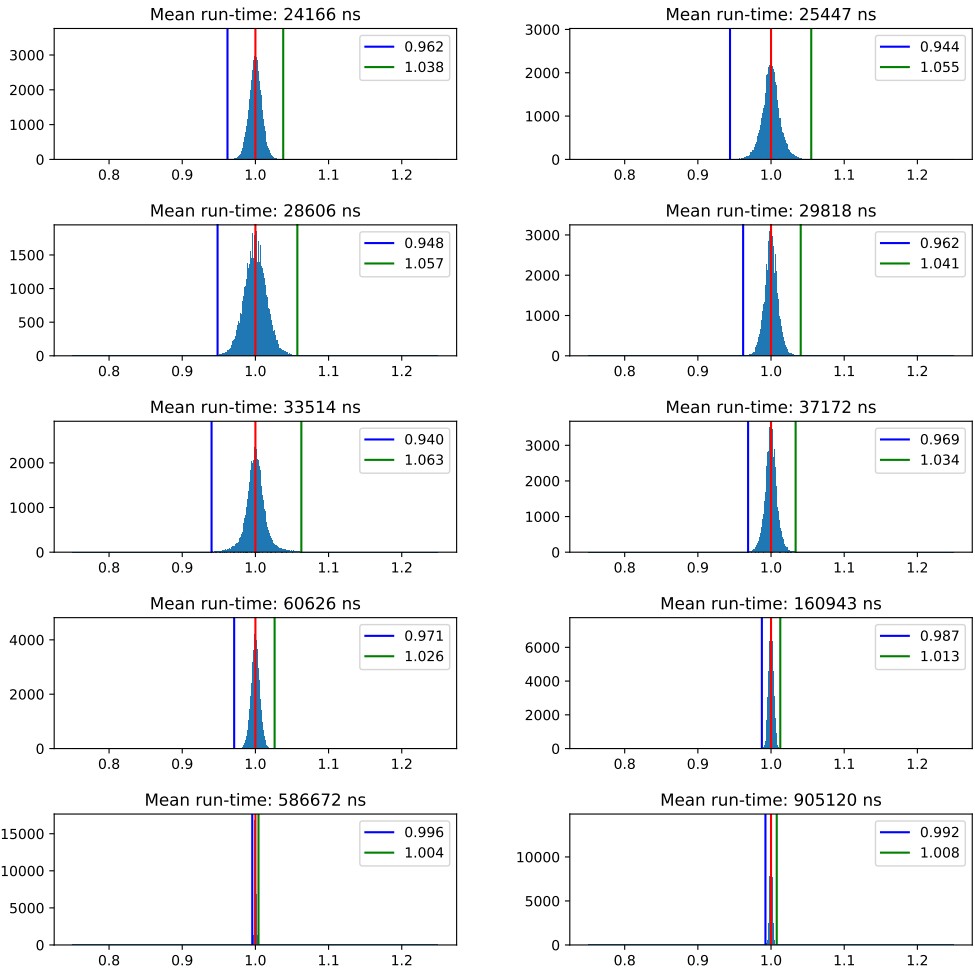

Figure 5: A profile showing the distribution of the runtime ratio noise of 10 different HLO graphs. The blue and green vertical lines encapsulate 99.9% of the ratios.

# E    PASS ANALYSIS GRAPH EXAMPLES

XLA Optimized Graph without HloConstantFolding Pass

XLA Optimized Graph

Figure 6: No HloConstantFolding pass runtime/XLA runtime = 0.51 (96.1% faster). This graph is too large to display its details, but the overall structure of both graphs can be seen to be visibly different. This is an example of how the removal of a simple pass like HloConstantFolding can cause compounding differences in the final result as the other passes/pipelines are applied.

# F  SEARCH OPTIMIZED VS XLA OPTIMIZED GRAPH EXAMPLES

## F.1  ALGEBRAIC SIMPLIFICATION

In these examples, the beam search and $f$-MCTS optimization strategies outperform XLA by removing specific graph rewrites that directly impact performance or allow for later passes to better optimize the graph.

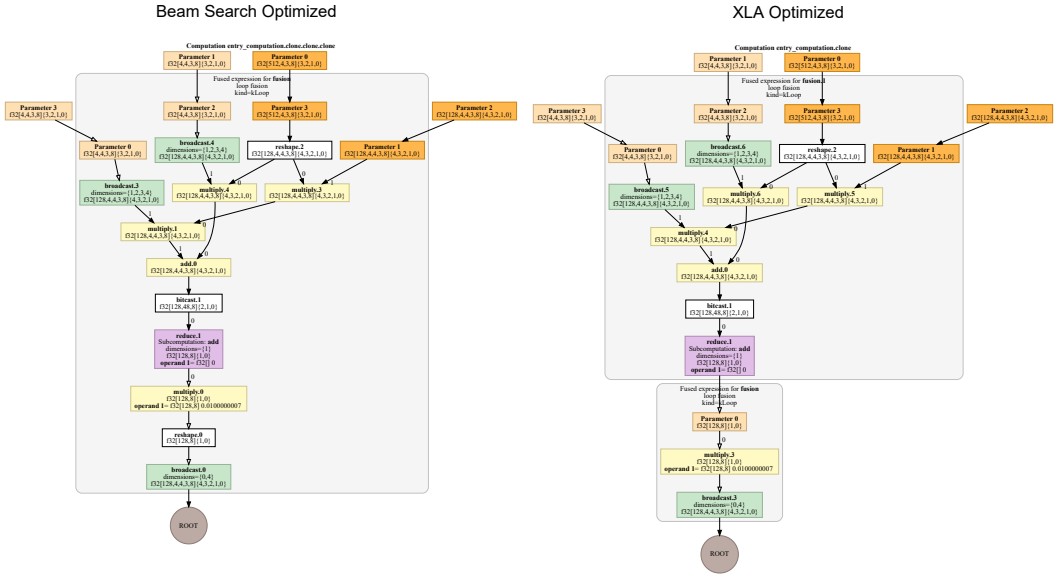

Figure 7: BS runtime/XLA runtime = 0.383 (161.1% faster). By not performing some optimizations during the Algebraic Simplification pass. The reshape instruction before the final broadcast instruction does not get optimized out, making later fusion passes able to fully fuse the HLO graph into one computation.

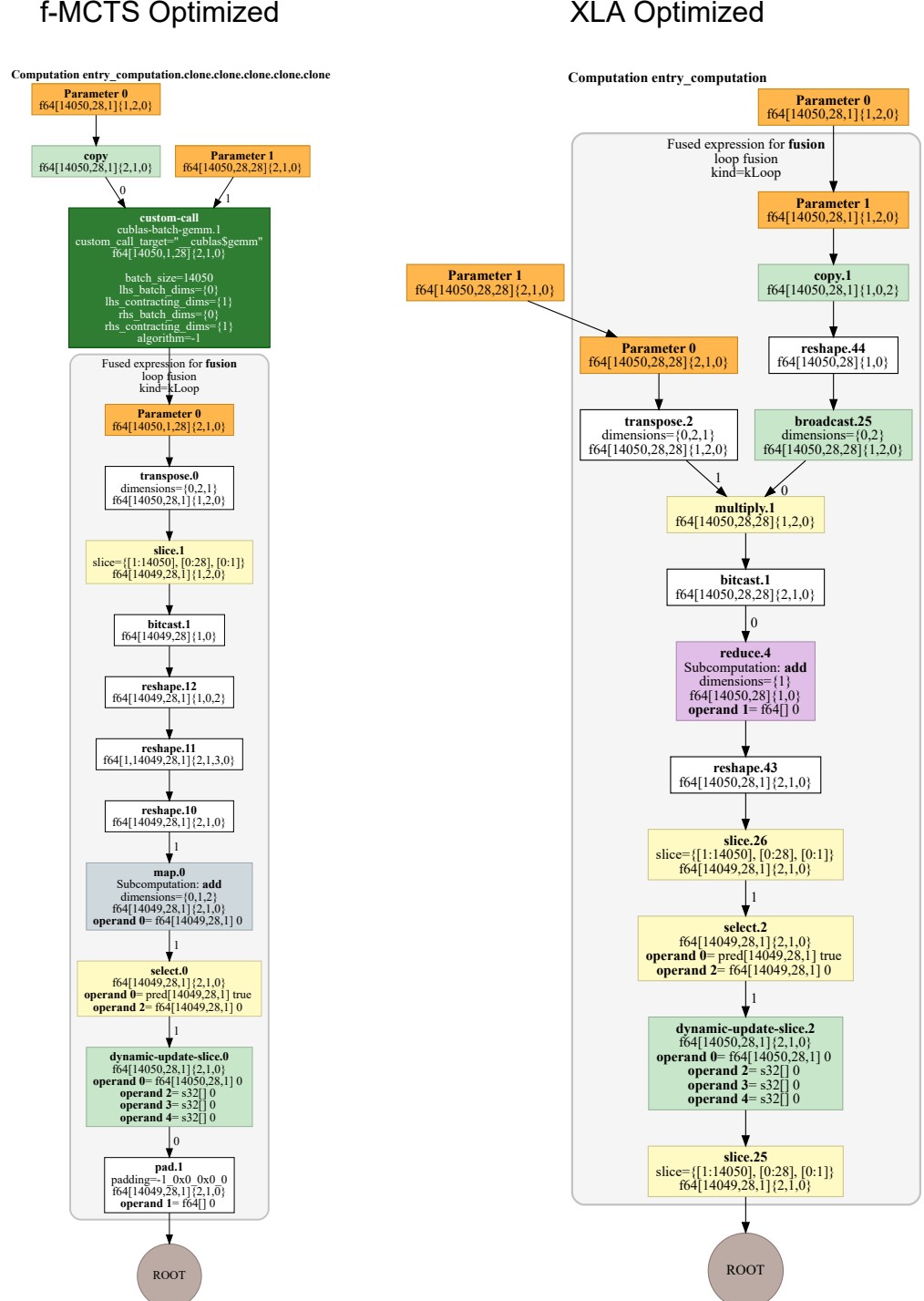

Figure 8: $f$-MCTS runtime/XLA runtime = 0.57 (75.4% faster). The $f$-MCTS optimized graph preserves the map instruction and batch-gemm custom call.

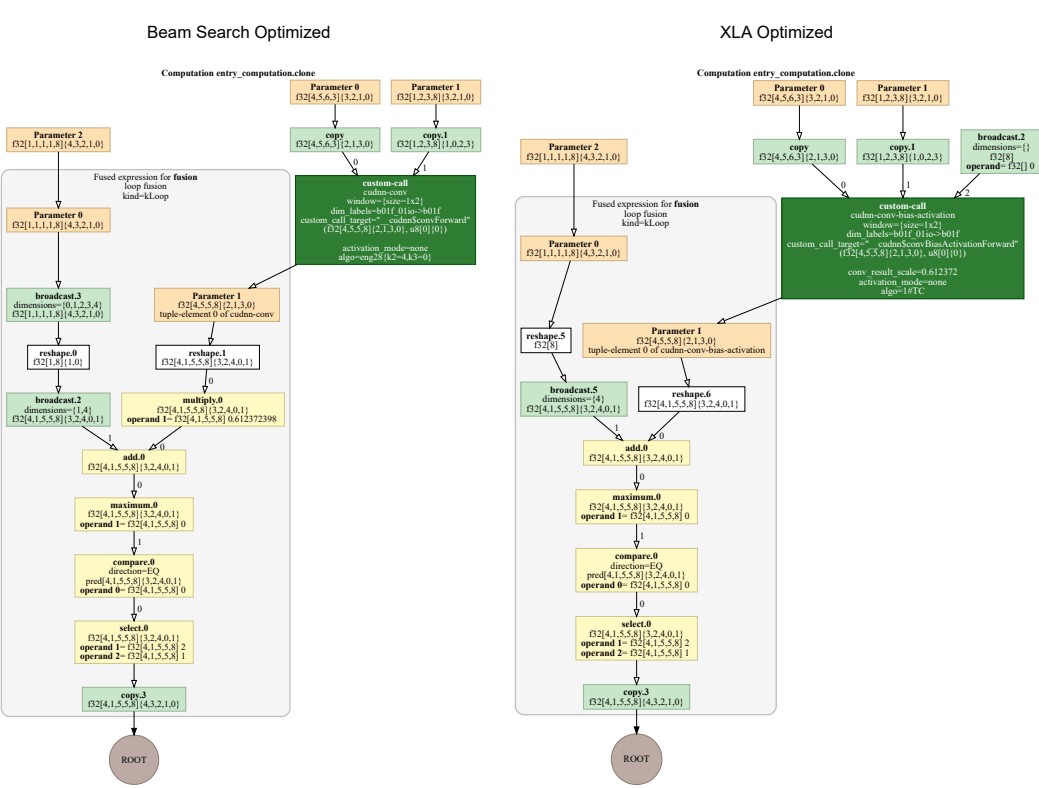

Figure 9: BS runtime/XLA runtime = 0.71 (40.8% faster). In this case, the beam search optimization results in a different set of instructions and custom calls.

Figure 10: BS runtime/XLA runtime = 0.79 (26.6 % faster). By not performing some optimizations during the Algebraic Simplification pass, later fusion passes can fully fuse the HLO graph into one computation.

### F.2 FUSION

### F.2.1 TRIVIAL EXAMPLES

In these examples, the beam search reduces the number of computations as compared to XLA by performing additional fusions of instructions and/or computations.

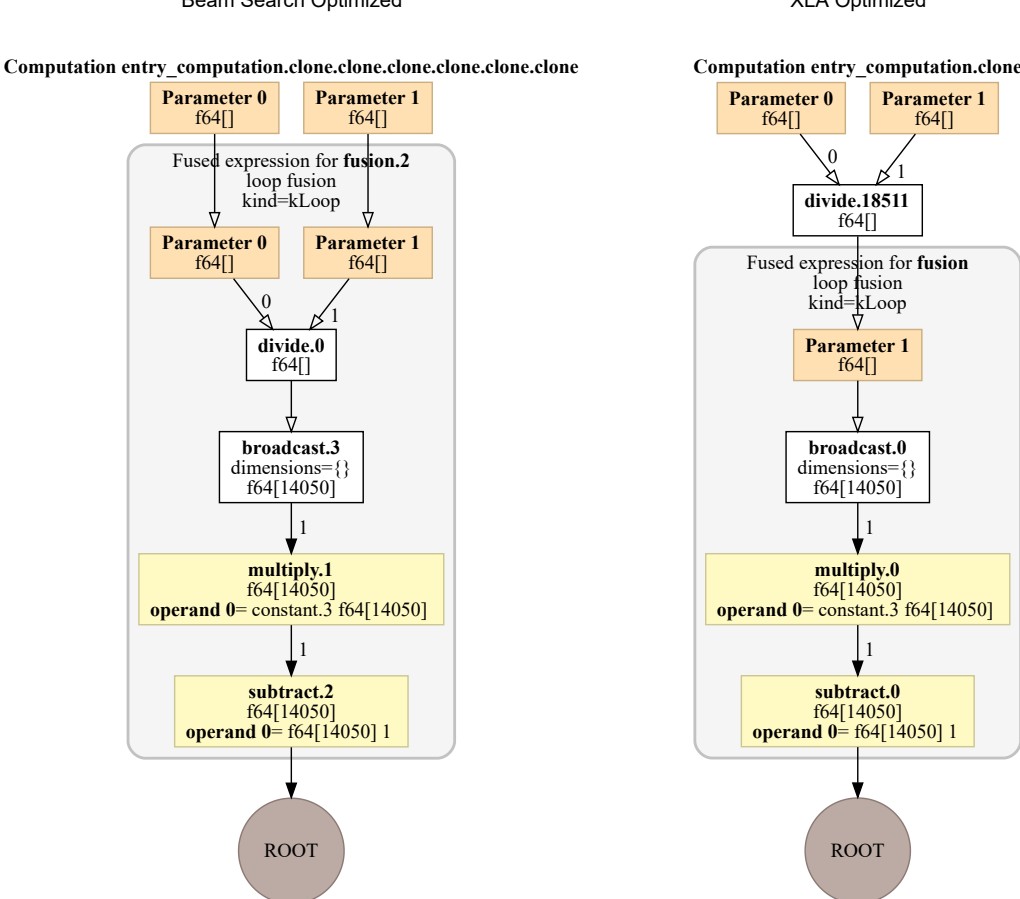

Figure 11: BS runtime/XLA runtime = 0.63 (58.7% faster). This is an example where the beam search finds a more optimized case that trivially involves just performing additional instruction and computation fusions.

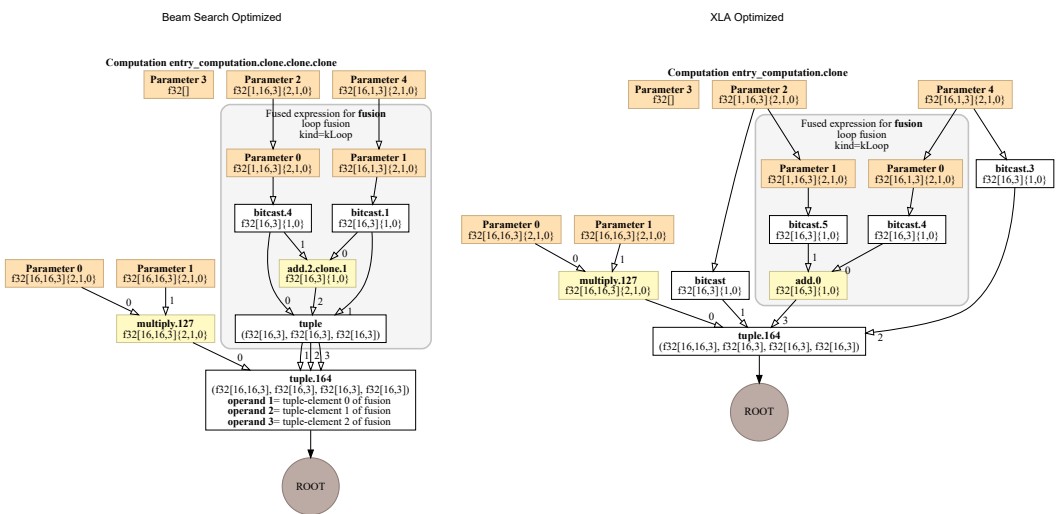

Figure 12: BS runtime/XLA runtime = 0.65 (53.8% faster). A second example is where the beam search finds a more optimized case that trivially involves just performing additional instruction and computation fusions.

### F.2.2 NON-TRIVIAL EXAMPLES

In these examples, the beam search and $f$-MCTS optimized graphs outperform the XLA graphs despite fusing fewer instructions/computations.

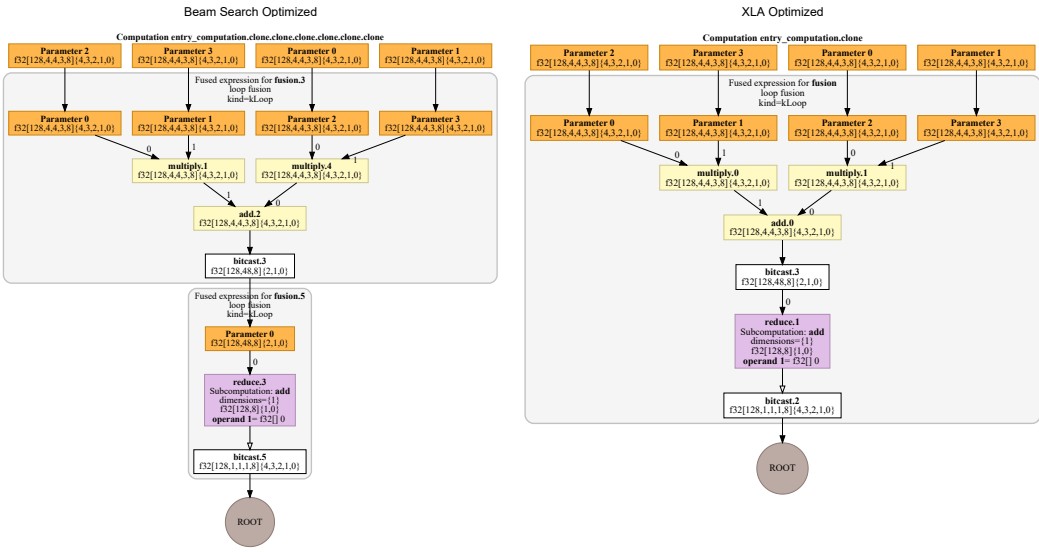

Figure 13: BS run-time/XLA run-time = 0.430 (132.5% faster).

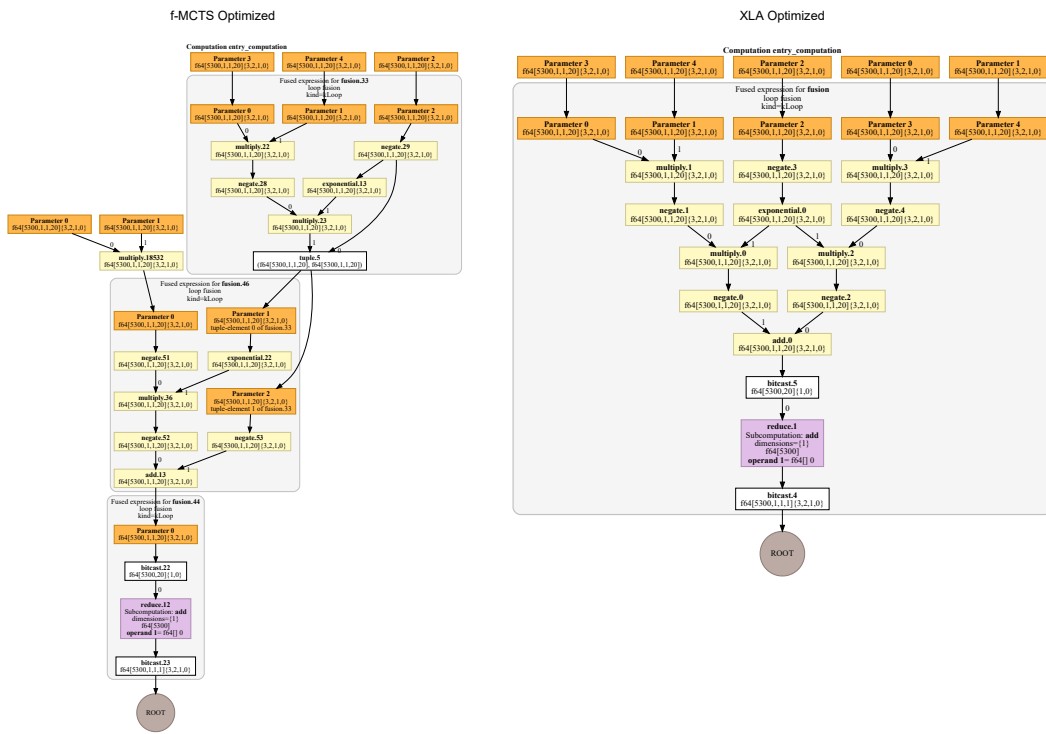

Figure 14: $f$-MCTS runtime/XLA runtime = 0.349 (186.5% faster).

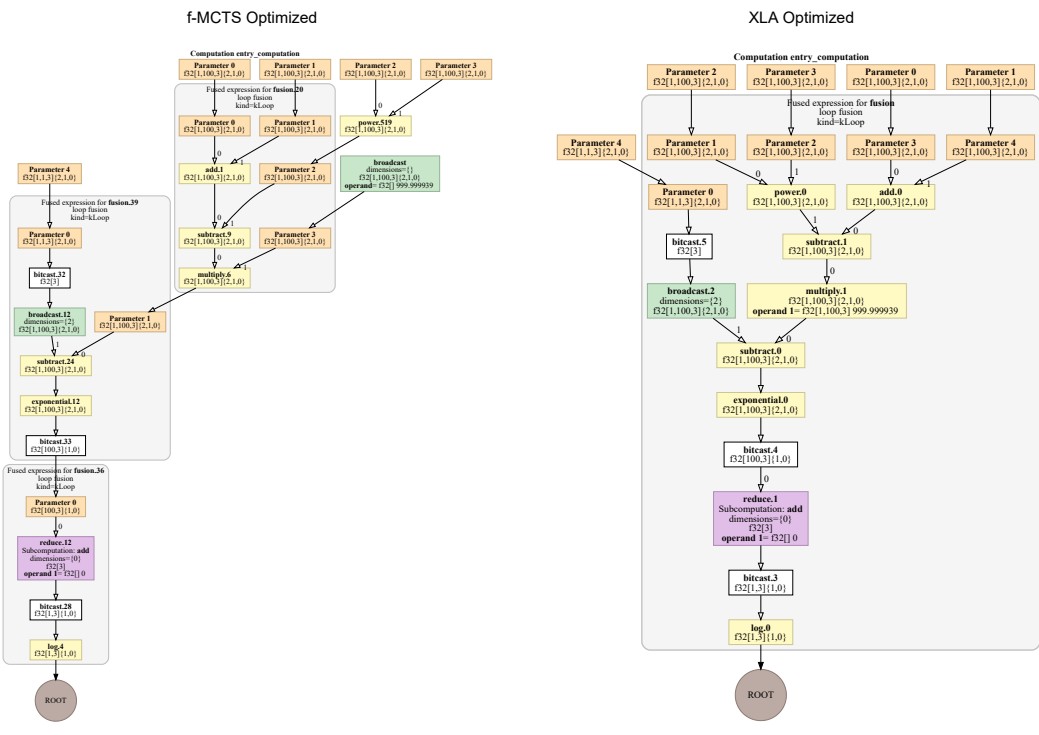

Figure 15: $f$-MCTS runtime/XLA runtime = 0.56 (78.6% faster).

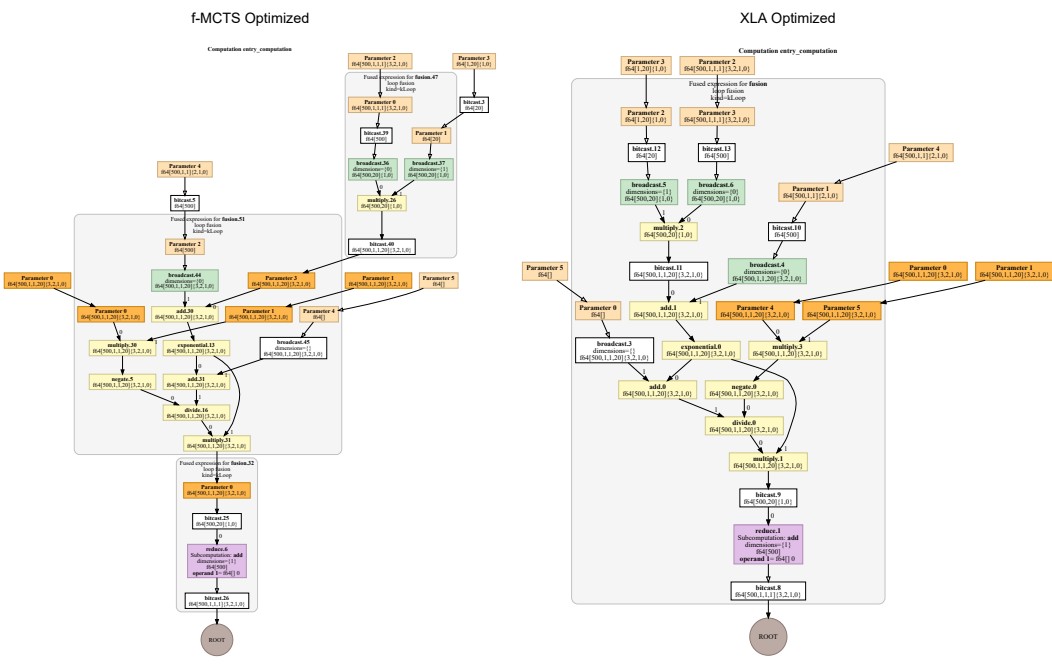

Figure 16: $f$-MCTS runtime/XLA runtime = 0.65 (53.8% faster).

f-MCTS Optimized

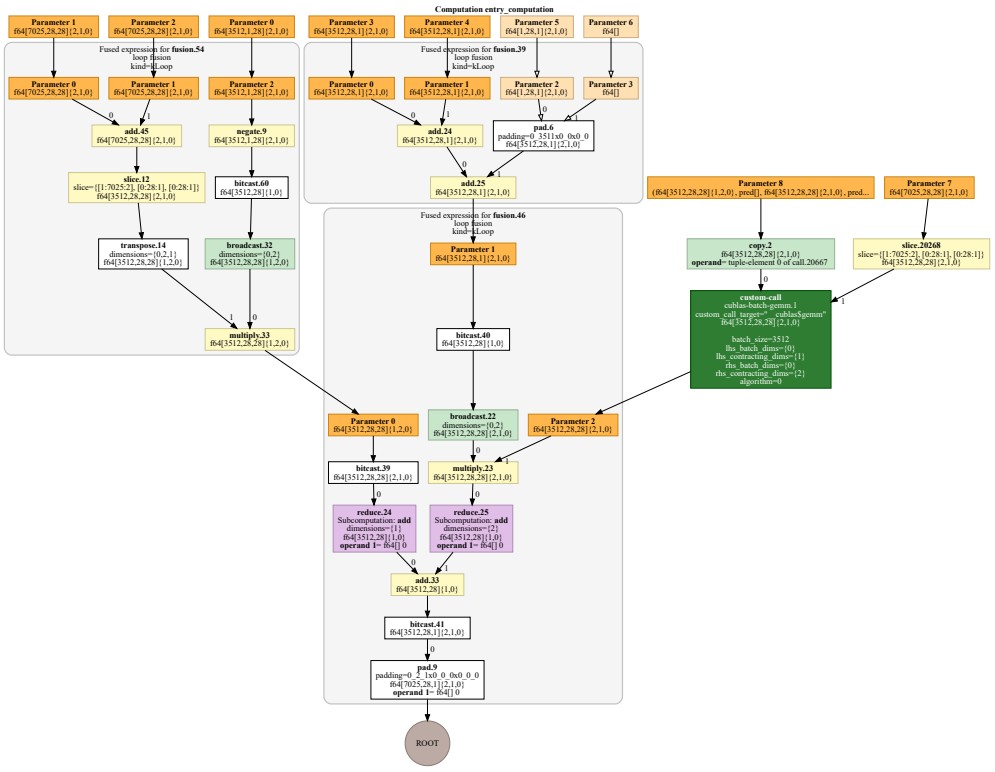

XLA Optimized

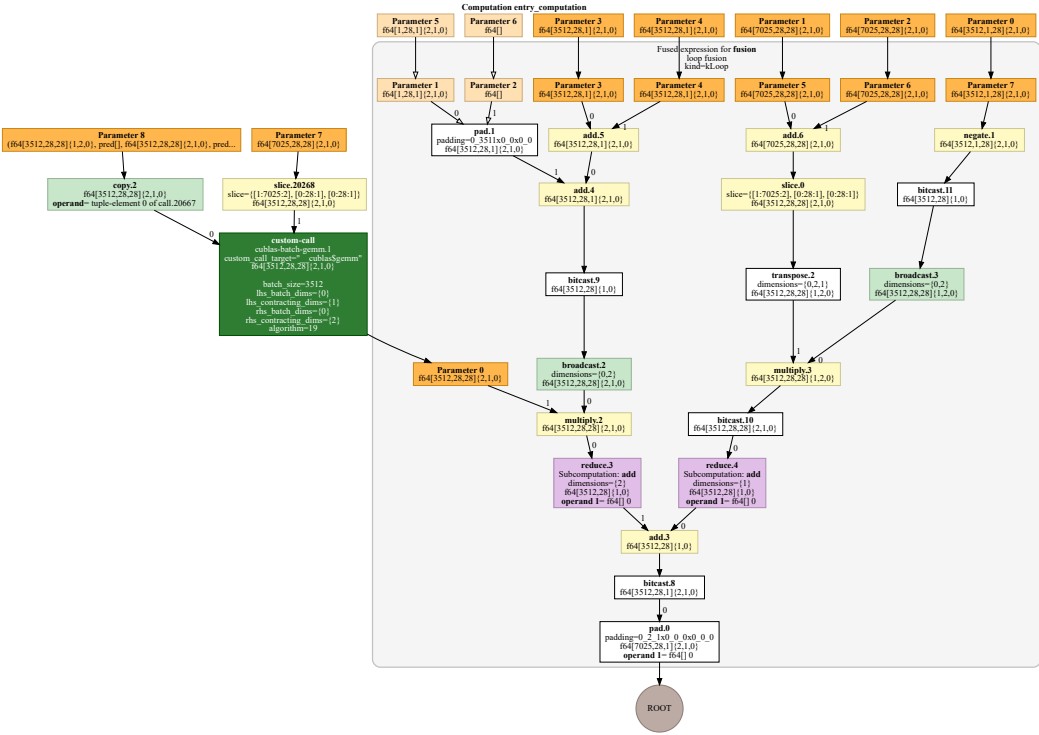

Figure 17: $f$-MCTS runtime/XLA runtime = 0.76 (31.6% faster).

