# OpenReview forum: "HloEnv: A Graph Rewrite Environment for Deep Learning Compiler Optimization Research"
_ICLR.cc/2023/Conference — Submitted to ICLR 2023_

### Official Review · Reviewer_fXG4 · 2022-10-30

**Confidence:** 4
**Correctness:** 3
**Technical Novelty And Significance:** 3
**Empirical Novelty And Significance:** 3
**Recommendation:** 6

**Clarity, Quality, Novelty And Reproducibility:**

+ Clearly written

- Possibly more improvement could be made to the evaluation (a little more analysis)
However, this should not deter the acceptance of the paper.


**Strength And Weaknesses:**

+ Deep learning compiler can have major impact on the community (training + inference performance)
+ Plans to open-source
+ Clearly explains key ideas and results.

- Only two examples which no clear evidence is provided about how good the baseline is.
  (while reviewer agrees that XLA is good baseline, but is it for all HW and for all networks?)

**Summary Of The Paper:**

Graph rewriting problem is an important problem in the field of deep learning compilers.
The paper notes that there is no good general testbed for deep learning compiler graph optimization research.
The paper develops and proposes a interface/framework for research.
With some preliminary results from the framework, the paper claims that it can provide more improvement compared to those of XLA.
The paper claims "We intend for HloEnv and our dataset to be an open-source, community-driven effort that helps spur advances in DL compiler optimization research."

**Summary Of The Review:**

The paper develops a framework for compiler optimization research, focusing on the graph rewrite problem.
The paper develops and proposes a interface/framework for research.
With some preliminary results from the framework, the paper claims that it can provide more improvement compared to those of XLA.
The paper claims "We intend for HloEnv and our dataset to be an open-source, community-driven effort that helps spur advances in DL compiler optimization research."

The paper is very well written and makes a very clear contribution in the field of compiler optimization research.
This should have large impact when fully panned out.
In fact, considering the fact that the authors promised to open-source the efforts, the paper should be a big asset to the community.

The evaluation is rather preliminary.
However, the main contribution of the paper comes from contributing a generalized framework for future development.
Even such preliminary results suffice to show the potential benefits of the work.

The paper would be a nice addition to the program!

---

### Official Review · Reviewer_UyBv · 2022-10-31

**Confidence:** 5
**Clarity, Quality, Novelty And Reproducibility:** See above.
**Correctness:** 3
**Technical Novelty And Significance:** 2
**Empirical Novelty And Significance:** 3
**Recommendation:** 3

**Strength And Weaknesses:**

Overall, I found this paper to be an interesting read and contain lots of interesting analysis about the XLA compiler. That being said, I do feel that there is a substantial amount of overlap with prior work done on the XLA compiler, and so I am somewhat uncertain about the contribution this paper provides over prior work.

### Prior work exploring learned heuristics over XLA graphs

For example, Paliwal et al. (2020) [1] provides a dataset of XLA graphs (although synthetic), Kaufman et al. (2021) [2]  explores a learned model for operator fusion, Yang et al. [3] explores equality saturation for graph rewrites, etc.

Thus, in my opinion, this statement in the abstract

> We show that using simple heuristics for decision-making can achieve on-par or better performance than XLA. Using search algorithms further boosts performance.

has already been substantially explored in prior works. So, if the authors wish to make a substantial contribution on this front, I'd have liked to see more comparison of the search approaches used in this paper compared to prior work. Alternately, adapting prior approaches to the paper's infrastructure would also be a great contribution.

### Dataset Quality
I also have some worries about the dataset provided in the paper. For one, I'm not sure what the primary advantages are of this benchmark compared to TenSet (cited in the paper). Second, I'm somewhat worries that the repos chosen to be included in this benchmark are not ... representative benchmarks. In particular, many of them look to be somewhat toy projects that are not used in the community. In my spot-check of the repos, it seemed like perhaps 2/3 (or more) were repos that had less than 100 stars, and some of them (like https://github.com/rasutt/Toy-neural-network-in-jax) were merely toy repositories.

I would not have an issue with the benchmarks chosen if they were merely to demonstrate performance improvements. However, as the dataset of this paper is meant to be a primary contribution, I think the inclusion of many low-traffic repositories is somewhat worrisome. (I would also have liked to see a split of where the graphs came from, since some of the repositories are mega-repositories like deepmind/deepmind-research). Benchmarks like MLPerf take great care to include "popular" and "representative" benchmarks, it worries me that, for example, I'm not sure I see a resnet implementation in any of the repositories provided.

More generally, I'm skeptical of approaches to benchmarks that provide a "bag of graphs". For a basic question, how many of these graphs are training benchmarks? How many of them are inference benchmarks? Although it's certainly *possible* to treat all of these graphs identically, the characteristics of inference graphs typically differ greatly from training graphs.

That being said, I do think the dataset improves meaningfully beyond existing XLA-HLO datasets, which range from proprietary to synthetic. However, the above concerns tamper my enthusiasm for this dataset.

### Conclusion

Overall, in my view, the main contributions listed by the paper are not strong enough for me to recommend acceptance. It has already been well established in the literature that XLA's heuristics can be improved through learned heuristics, and in my view the dataset contribution suffers from 1. lack of widely-used models, and 2. (related to 1) lack of more insight into the graphs and what they represent.

Thus, I think the primary contributions of this paper are: 1. A great investigation into what XLA heuristics matter, and 2. released code/data (as opposed to prior work which was mostly closed AFAIK). However, I think neither of these contributions are sufficiently technically novel, and I have several worries about the quality of the dataset.

[1] https://arxiv.org/pdf/1905.02494.pdf

[2] https://proceedings.mlsys.org/paper/2021/file/85d8ce590ad8981ca2c8286f79f59954-Paper.pdf

[3] https://proceedings.mlsys.org/paper/2021/hash/65ded5353c5ee48d0b7d48c591b8f430-Abstract.html

**Summary Of The Paper:**

This paper provides 1. a dataset of XLA-HLO graphs, 2. an environment to work with XLA in (HLOEnv), and 3. they explore the XLA optimization passes.

The dataset is gathered from 26 Jax repos, of varying sizes.

HLOEnv provides 2 main additions beyond the basics: 1. A dry-mode for many of XLA's optimizations (i.e. the ability run the optimizations and collect information without modifying the graph), and 2. an alternate graph representation that provides an "alternate graph" representation of the optimization opportunity.

They also explore the optimizations done by XLA, and find two main optimizations that have a significant impact on performance - algebraic simplification and operator fusion. They explore various heuristics and find that it's possible to improve upon XLA's existing heuristics, even with fairly simple choices.



**Summary Of The Review:**

See Conclusion.

---

### Official Review · Reviewer_ZQ7g · 2022-11-04

**Confidence:** 4
**Correctness:** 2
**Technical Novelty And Significance:** 2
**Empirical Novelty And Significance:** 2
**Recommendation:** 3

**Clarity, Quality, Novelty And Reproducibility:**

Novelty
--------
There is a lot of very similar work that is not cited in this paper. The paper presents  graph format that introduces alternatives nodes and chooses between them. This representation is nearly identical to e-graphs (https://egraphs-good.github.io) which has a lot of literature published that is not cited here. Some of that literature is even looking at deep learning (https://arxiv.org/pdf/2101.01332.pdf). Similar tree search has been used in space as well (https://halide-lang.org/papers/autoscheduler2019.html) but was not cited.  This paper needs to clarify its novelty considering all of these publications that tackle very similar problems.

Quality
--------

Results are reported in averages that make it hard to understand the performance. For instance, it would be much stronger to show a histogram of timings for the several thousand graphs rather than a single averaged number.

The results presented as averages are themselves pretty weak, with both optimizations only showing improvements in the 1-2% range. At this level, it might just be noise.

Reproducibility
-----------------
If the code is open sourced, this would be reproducible. Without it, it would be difficult to figure out where in XLA's rewrites was modified to extract rewrites, or how to filter XLA graphs to make them safe to use with the algebraic rewrite pass.


**Strength And Weaknesses:**

Strengths
-----------
A fairly large dataset of XLA graphs as a benchmark for experimenting with optimizers is useful.

The description of how the system works was fairly easy to follow.

Weaknesses
--------------

> Of the 94332 inst-10-20 sub-graphs generated, we filtered out 34000 graphs for which the correctness of the graph was not dependent on any graph rewrites performed by the Algebraic Simplification pass.

The approach of extracting rewrite rules from existing passing seems pretty fragile if ~2/3rds of the graphs produce incorrect results when it was changed. Similarly the fusion pass seems to filter out a similar proportion of the graphs because the method of searching would take too long to search. The paper needs to make this limitation much more clear, it is only mentioned in an appendix. Furthermore, it is not really a fair comparison to baseline XLA to exclude graphs it would compile fine. If averages are being reported, I would expect that the approach in the paper would have to include the numbers for the no-speedup cases where it cannot run, or would take too long to run.

The datasets are subsets of real world programs. Measuring performance improvements on a subset of a program is not necessarily representative of the whole program because time is not evenly spent in each operator.  I'd expect the best-improved subsets of programs to be much more improved than the program overall because maybe the subset chosen was precisely that part that would speed up a lot.

**Summary Of The Paper:**

The paper presents a system for extracting XLA graphs from the XLA compiler with annotations where the XLA compiler would have performed a rewrite that preserve the graph before and after the rewrite as alternatives. It looks into a system that applies different decisions that the default XLA rewrites by using beam search or monte-carlo tree search to choose between alternatives. It evaluates this system on a dataset of XLA programs gathered from GitHub repos.

**Summary Of The Review:**

There are a number of factors here that suggest this paper needs more work before publication. It does not cite work like e-graphs which are very close to the representation used in the paper. Its results are also pretty weak, with only 1-2% overall speedups. Maybe this is due to averaging a lot of programs that are not effected by the optimizations, but the results are not presented in a way that makes it clear how the results would look if run on end-to-end programs, or whether improvements are due to a couple outliers.

---

### Decision · Program_Chairs · 2023-01-20

**Decision:**

Reject

**Justification For Why Not Higher Score:**

The reviewers raise several important objections that mean that to publish the paper in its current form would not be a service to the community.  After addressing the reviewers' concerns, this will be a valuable paper, but the changes required are sufficiently large to require re-review.

**Justification For Why Not Lower Score:**

n/a

**Metareview: Summary, Strengths And Weaknesses:**


The reviewers make several suggestions that indicate there is potential value in this dataset and environment, but that more work needs to be done to make the paper ready for publication.  For example, on the comparison to e-graphs, it really will be necessary to perform a quite fine-grained comparison between the paper's proposal and e-graphs.  It may even be that the paper is improved by switching to the e-graph codebase and representation.

The reviewers ask for more careful curation of the source graphs, and as AC I tend to agree - taking resnet for example, there are numerous non-mlperf implementations; it would make sense to follow the reviewers suggestions there.  R2 would like a more "explicit split of where the graphs came from" - again this appears a reasonable suggestion.

In all, the summary is that this paper will likely have higher impact if rewritten and re-reviewed.